# Seasonal Influenza Vaccination Programs in the Americas: A Platform for Sustainable Life-Course Immunization and Its Role for Pandemic Preparedness and Response

**DOI:** 10.3390/vaccines12121415

**Published:** 2024-12-16

**Authors:** Francisco Nogareda, Margherita Ghiselli, Martha Velandia-González, Bremen de Mucio, Jorge Jara, Paula Couto, Angel Rodriguez, Marc Rondy, Andrea Vicari, Murat Hakan Ozturk, Shoshanna Goldin, Alba Vilajeliu, Eva Leidman, Jaymin Patel, Julie Carlton, Ashley L. Fowlkes, Eduardo Azziz-Baumgartner, Daniel Salas Peraza, Alba Maria Ropero

**Affiliations:** 1Special Program Comprehensive Immunization, Pan American Health Organization (PAHO)/World Health Organization (WHO), Washington, DC 20037, USA; velandiam@paho.org (M.V.-G.); jarajor@paho.org (J.J.); salasdan@paho.org (D.S.P.); 2Women and Reproductive Health, Latin American Center for Perinatology, Pan American Health Organization (PAHO)/World Health Organization (WHO), Washington, DC 20037, USA; demuciob@paho.org; 3Health Emergencies Department, Pan American Health Organization (PAHO)/World Health Organization (WHO), Washington, DC 20037, USA; coutopau@paho.org (P.C.); rodrigueza@paho.org (A.R.); rondymar@paho.org (M.R.); vicarian@paho.org (A.V.); 4Revolving Fund for Access to Vaccines, Pan American Health Organization (PAHO)/World Health Organization (WHO), Washington, DC 20037, USA; ozturkmur@paho.org; 5World Health Emergencies Program, World Health Organization (WHO), 1211 Geneva, Switzerland; goldins@who.int; 6Immunization, Vaccines and Biologicals, World Health Organization (WHO), 1211 Geneva, Switzerland; avilajeliu@who.int; 7Coronavirus and Other Respiratory Viruses Division, National Center for Immunization and Respiratory Diseases, US Centers for Disease Control and Prevention, Atlanta, GA 30333, USA; wzu0@cdc.gov; 8Global Immunization Division, Global Health Center, US Centers for Disease Control and Prevention, Atlanta, GA 30333, USA; isr0@cdc.gov (J.P.); wvd9@cdc.gov (J.C.); 9Influenza Division, National Center for Immunization and Respiratory Diseases, US Centers for Disease Control and Prevention, Atlanta, GA 30333, USA; ahl4@cdc.gov (A.L.F.); eha9@cdc.gov (E.A.-B.); 10Country Office for the Dominican Republic, Pan American Health Organization (PAHO)/World Health Organization (WHO), Ensanche La Fé, Santo Domingo 10514, Dominican Republic; roperoal@paho.org

**Keywords:** influenza, COVID-19, respiratory syncytial virus, vaccines, immunization, outbreaks, maternal vaccination, vaccination week in the Americas, pandemic preparedness and response, life-course immunization

## Abstract

**Background:** Vaccination is one of the most effective measures to prevent influenza illness and its complications. Since the 1980s, countries and territories in the Americas have progressively implemented influenza vaccination operations in high-risk priority groups—such as older adults, pregnant persons, persons with comorbidities and health workers. **Methods:** In this review, we present the history and progress of the seasonal influenza program in the Americas, how the program contributed to the efficient and timely roll-out of the COVID-19 vaccines during the pandemic, and how the program can be used to promote immunization operations across the life span for existing and future vaccines. **Results:** The influenza A(H1N1)pdm09 pandemic in 2009 and the COVID-19 pandemic in 2020–2023 underscored the importance of having a robust seasonal influenza vaccination program for pandemic preparedness and response. Overall, countries with existing seasonal influenza vaccination programs were better prepared and rolled out the delivery of COVID-19 vaccines more quickly and effectively compared to other countries where the influenza vaccination platform was weak or non-existent. **Conclusions:** Traditionally, national immunization programs of developing countries have been predominately focused on newborns, children younger than five years and school-aged children while often limiting their investment in effective adult vaccination programs; these programs are typically isolated to high-income countries. Countries in Latin America have been the exception, with strong influenza vaccination programs for adults regardless of national income level. The presence of functional and effective adult influenza vaccination programs can also facilitate the acceptance and uptake of other adult vaccines targeting priority groups at higher risk for severe illness or complications.

## 1. Introduction

Seasonal influenza epidemics pose a substantial burden of disease, affecting individuals of all ages and causing substantial morbidity and mortality. The World Health Organization (WHO) estimates that each year influenza causes 1 billion infections, including 3–5 million severe illnesses and 290,000–650,000 influenza-related deaths globally [1]. In the Americas, data reported from member states to SARI*net* plus—a Pan American Health Organization (PAHO) integrated respiratory illness surveillance network created in 2014 to monitor influenza and other respiratory viruses in the Americas—indicate that influenza contributes significantly to the respiratory disease burden in the Region [2]. It is estimated that between 716,000 and 829,000 influenza-associated respiratory hospitalizations and 41,007 to 71,710 influenza-associated deaths occur every year in the Region [3,4]. Preliminary findings from a 2022–2023 analysis covering 70% of the population, estimate approximately 55 million annual cases of mild influenza in the Americas. In 2020, SARI*net* plus was expanded to integrate COVID-19 surveillance into existing sentinel-based severe acute respiratory infection (SARI)/influenza-like illness (ILI) surveillance systems [5,6].

Vaccination is one of the most effective measures to prevent influenza illness and its complications. Safe and effective influenza vaccines are available and are routinely used globally [7]. Historically, the Americas has been one of the regions with the highest seasonal influenza vaccination coverage in the world [8], yet some countries struggled to sustain routine immunization services during the COVID-19 pandemic [9]. The first seasonal influenza vaccine was developed in the 1930s and introduced in the United States in 1945. Since 1980, countries and territories in the Americas have been progressively implementing and expanding influenza vaccination activities in priority groups—such as older adults, pregnant persons, children 6–23 months of age, persons with comorbidities and health workers—and integrating this vaccine into their national immunization programs [8,10,11]. The influenza A(H1N1)pdm09 pandemic in 2009 and the COVID-19 pandemic in 2020–2023 underscored the importance of having a robust influenza vaccination program for pandemic preparedness and response [12,13]. Countries with existing seasonal influenza vaccination programs targeting adult populations were better prepared and rolled out the COVID-19 vaccines more quickly in comparison to other countries where the influenza vaccination program was weak or non-existent [14,15,16]. Also, high-performing influenza vaccination programs have been shown to facilitate the acceptance and uptake of other vaccines among high-risk priority groups, such as pregnant persons, health workers and older adults [14,17,18].

In this review, we describe the progress of the seasonal influenza vaccination in the Americas and the lessons learned about building robust, sustainable life-course approach immunization programs and assess their role in facilitating the rollout of novel or existing vaccines during public health emergencies. Its purpose is to describe the key lessons learned about sustainable vaccine introduction, as well as examine how the program contributed to a timely roll-out of COVID-19 vaccines during the recent pandemic. We discuss how these experiences can serve as a blueprint for the introduction of new vaccines—for example, against the respiratory syncytial virus (RSV) or during preparedness and response operations to new epidemics and pandemics—for similar adult priority groups. We intend this review to be useful in delineating the next steps for advancing influenza vaccination programs in the Americas and in supporting other WHO regions in improving their own programs to reduce the burden of influenza morbidity and mortality globally through a life-course immunization approach.

## 2. Influenza Vaccination Program in the Americas

### 2.1. Influenza Vaccine Introduction

Seasonal influenza vaccination in the Americas was first introduced in the United States in 1945 [19] and in Canada and Chile in 1980 with a focus on older adults. Since then, countries in the Americas have been progressively introducing the vaccine to risk groups and expanding the vaccination activities to other priority groups. While in 2002 only seven countries in the Americas were vaccinating priority groups, in 2023, 41 (93%) out of the 44 countries and territories that report their data through the PAHO/WHO and the United Nations Children’s Fund (UNICEF) electronic Joint Reporting Form (eJRF) were using the vaccine in at least one priority group. Only Guyana, Haiti and Saint Vincent and the Grenadines do not have a seasonal influenza vaccination program targeting any priority group. Older adults and health workers were targeted for vaccination in 40 (98%) countries and territories, persons with chronic diseases in 36 (88%), pregnant persons in 35 (85%), and children older than 6 months in 30 (73%) countries [8,10,11] (Table 1 and Figure 1).

### 2.2. Influenza Vaccine Formulations

Seasonal influenza vaccine composition is updated every year for both hemispheres to optimize the effectiveness of the vaccine with the circulating strains. In 2023, 27 (66%) countries/territories with a seasonal influenza vaccination program used the Northern Hemisphere vaccine formulation, while 14 (34%) used the Southern Hemisphere vaccine formulation (Table 1) [8]. To date, no country/territory in Latin America and the Caribbean has reported the public-sector use of live attenuated influenza vaccines (LAIV) to PAHO/WHO. These countries use only inactivated influenza vaccines. However, LAIV has been used in Canada for individuals aged 2–59 years [20] and in the United States for healthy non-pregnant people aged 2–49 years [21]. Since 2024, most of the countries in the Region are using trivalent vaccines (composed of two influenza A strains and one influenza B) following the recommendations of the WHO influenza vaccine composition advisory committee [22].

### 2.3. Procurement Through the PAHO Revolving Fund for Vaccines in the Americas

While Argentina, Brazil, Canada, Curaçao, Chile, Mexico, Sint Marteen, USA, and Venezuela produce or purchase influenza vaccines directly from manufacturers, 32 of the 41 (78%) countries/territories with seasonal influenza vaccination programs access influenza vaccines through the PAHO Revolving Fund for Access to Vaccines (RFV) [23]. This technical cooperation mechanism provides timely access to quality-assured vaccines and immunization supplies at affordable prices for countries and territories in Latin America and the Caribbean. By consolidating requirements from National Immunization Programs and orchestrating pooled procurement, the RFV leverages economies of scale to improve member states’ purchasing power and supports all procurement and logistics processes. The RFV has been instrumental in enabling rapid and equitable access to new vaccines and in supporting regional goals for disease elimination. Annually, more than 30 million doses of seasonal influenza vaccines are procured through the RFV mechanism. The RFV facilitated the distribution of vaccines for routine immunization programs and emergencies, including COVID-19 vaccines during the pandemic in collaboration with the COVAX facility.

### 2.4. Quality of Influenza Vaccination Monitoring Data

Despite significant improvements in influenza vaccination program efforts in recent years, the completeness and timeliness of influenza vaccination data remain suboptimal in many countries across the Region [8,11]. First, countries often rely on outdated census data, health workforce statistics, or rough estimations to determine the size of different target groups, such as pregnant persons or people with chronic conditions. Also, countries in Latin America and the Caribbean inconsistently report the number of vaccinated persons per season to PAHO—leaving gaps in the regional estimates of disease burden and vaccination coverage rates. Moreover, merging data from different sources—such as national vaccination registries, healthcare provider registers and surveys—can be challenging due to variations in data formats, definitions and collection methods within the same country. Finally, the demand for vaccination fluctuates each year, complicating efforts to monitor instances of stockout. On the other hand, regional-level information systems are designed to support interoperability between various systems. For example, great strides have been made in facilitating the joint analysis of seasonal influenza surveillance information systems and vaccination registration platforms to calculate regional vaccine effectiveness estimates.

### 2.5. Seasonal Influenza Vaccination Coverage by Priority Group

It is estimated that every year, more than 100 million doses of influenza vaccine are administered in the Americas using different strategies (e.g., primary care and health-based immunization, door-to-door outreach activities, vaccination campaigns). Target groups and eligible age groups considered for vaccination (e.g., older adults, people with comorbidities and children) vary across countries, which makes comparisons difficult. For instance, some countries target older adults aged 60 years and older, while others focus on persons aged 65 years or older. Countries and territories systematically report data on influenza vaccination programs and vaccination coverage on an annual basis to PAHO, WHO and the United Nations Children’s Fund (UNICEF) via the eJRF [8]. These data are then published annually in publicly accessible “Immunization in the Americas” brochures [11]. While the reported PAHO influenza vaccine coverage rate is the highest by any WHO Region, the Americas still fall short of global goals. According to the 2023 eJRF report, only Colombia, Cuba, Dominican Republic and Mexico achieved the 75% target coverage rate set by WHO for older adults. Regional median vaccination coverage achieved in 2023 among high-risk priority groups in the Region are presented in Table 2 [21].

### 2.6. Vaccination Week in the Americas

Vaccination Week in the Americas (VWA) is a yearly regional initiative that promotes the benefits of vaccination to all persons in the Region. In its 22-year history, more than 40 countries and territories reached 1.15 billion people during the VWA, according to country-level tallies compiled by PAHO. The initiative aims to close the most urgent immunity gaps and contribute to disease elimination efforts in the Americas, all while offering versatile plans for operational execution, communication, and political advocacy that can be adapted to each country’s priorities.

To date, more than 502 million doses of influenza vaccine have been delivered through this initiative across the Americas, with 47 million doses in 2023 alone [24]. Many countries in the Southern Hemisphere use the VWA platform to kick off their seasonal influenza vaccination activities. Also, between 2021 and 2023, approximately 150 million COVID-19 vaccine doses were administered across the Region under the umbrella of the VWA.

### 2.7. Vaccine Effectiveness and Impact Evaluation of Influenza Vaccination

Established in 2013, the Network for the Evaluation of the Effectiveness of the Vaccine in Latin America and the Caribbean-influenza (REVELAC-I) uses the test-negative design approach to estimate the vaccine effectiveness of influenza vaccines [25]. It leverages sentinel surveillance systems for severe acute respiratory infections (SARIs) at the country level to produce semiannual regional and national estimates of influenza vaccine effectiveness. It also evaluates the impact of the vaccination expressed as the number of influenza illnesses averted through vaccination. Such estimates can be used as inputs for economic analyses to evaluate the value of currently licensed and future vaccines to prevent viral respiratory illnesses. This well-established and flexible platform was expanded to estimate the effectiveness of other respiratory viruses’ vaccines, including those to prevent COVID-19. For the 2024 season, five countries (Argentina, Brazil, Chile, Paraguay, Uruguay) are collecting and analyzing SARI sentinel surveillance and immunization data to generate influenza and COVID-19 vaccines effectiveness against hospitalization. Estimates are stratified by vaccine type, age group and by circulating subtype of influenza virus and variant of SARS-CoV-2 [26]. Interim 2024 vaccine effectiveness estimates against influenza-associated hospitalization are 34.5% (95% confidence interval (CI): 26.4, 41.7) for any type of influenza, 37.2% (95% CI: 22.0, 49.5) for influenza A(H1N1)pdm09 and 36.5% (95% CI: 25.6, 45.7) for influenza A(H3N2) [27]. Mid-season and end-of-season influenza vaccine effectiveness estimates are reported twice a year to the Global Influenza Vaccine Effectiveness (GIVE) working group and presented at the meeting on the Composition of Influenza Virus Vaccines contributing to the recommendation for the viral composition of the next season’s influenza vaccine. Since January 2024, the network has been expanded further to generate effectiveness estimates on preventing severe disease in infants for the maternal RSV vaccine and for the infant receipt of RSV monoclonal antibodies (i.e., laboratory-made proteins that are administered to newborns to protect them against severe RSV disease).

### 2.8. Partnerships

PAHO maintains strong partnerships with the Ministries and Departments of Health of all 51 countries and territories of the Americas. In the context of influenza vaccination, PAHO is working closely with most member states to maintain the high performance of their existing program or prepare for the introduction of the seasonal influenza vaccine in the national immunization schedule. This close collaboration is essential for the ongoing collection and collation of complete, timely and reliable surveillance data on SARI and ILI events, which in turn allows the Region to assess viral circulation in both hemispheres and take appropriate action to prevent epidemics. Also, PAHO is in a technical and financial partnership with the U.S. Centers for Disease Control and Prevention (CDC), which has been a supporter of the SARInet and REVELAC-I networks since their inception.

## 3. Leveraging Seasonal Influenza Vaccination Programs for Pandemic Vaccination in the Americas

### 3.1. Seasonal Influenza Vaccination for Pandemic Preparedness and Response

The existence of seasonal influenza vaccination programs contributed to the response to both the 2009 influenza A(H1N1)pdm09 pandemic and the COVID-19 pandemic. An analysis of the 2009 pandemic revealed that countries with a pre-existing seasonal influenza vaccination program were able to more quickly roll-out the pandemic influenza A(H1N1)pdm09 vaccine [12]. Similarly, 12 months after the introduction of COVID-19 vaccines, countries with an influenza vaccination program had reached an average vaccination coverage of 47% of the total population with the primary series compared to 22% in countries without influenza programs [14]. During and after the COVID-19 pandemic, several assessments suggested that, among the potential predictors of COVID-19 vaccine introduction and roll-out success (i.e., national maturity levels for childhood, adolescent, and adult vaccinations; experience conducting emergency vaccination; tendency to be early adopters of new vaccines; degree of trust in the vaccine product; availability of human resources; income status) [14,15], only the presence of seasonal influenza vaccination programs were associated with a swift introduction capacity for COVID-19 vaccines and higher coverage levels [13]. These results are compatible with findings from COVID-19 vaccine post-introduction evaluations (c-PIE) conducted in 20 countries globally between 2021 and 2022 and supported by the findings of key-informant interviews with the WHO regional and country-level immunization officers [14,16]. The following factors were identified as essential for the successful introduction of the COVID-19 vaccines, according to the WHO technical guidelines for the National Deployment and Vaccination Plan (NDVP) [28].

### 3.2. Regulatory Pathways for Authorization, Vaccine Introduction and Roll-Out

Countries with seasonal influenza vaccination programs had more straightforward regulatory approvals or emergency authorization mechanisms for the pandemic COVID-19 vaccines and established procedures for importation and use in the target population. Data collected through standardized COVID-19 post-introduction evaluation conducted at national and subnational levels in LMIC indicated that 29% of countries with influenza vaccination programs encountered barriers to regulatory approval of COVID-19 vaccines compared to 40% of countries without an influenza vaccination program [14]. On the other hand, countries without influenza vaccination programs were often slower in their operations for vaccine introduction, distribution, and administration.

### 3.3. Vaccination in Priority Groups

Existing seasonal influenza vaccination strategies targeting priority groups in the adult population (i.e., older adults, health workers, pregnant persons, and adults with chronic conditions) played a crucial role in guiding effective operations and achieving high vaccination coverage of COVID-19 vaccines within these populations. For example, the COVID-19 vaccination coverage of at least one dose among health workers was 83% in countries with an influenza vaccination program compared to 69% in countries without health worker vaccination programs. Also, adults with comorbidities had higher COVID-19 vaccine uptake in countries with health worker vaccination programs than in countries without health worker vaccination programs [14]. In addition, influenza vaccination programs for older adults provided sources of data (e.g., nominal electronic immunization registries, collaboration with nursing homes and medical specialists) to identify and locate this population at the time of COVID-19 vaccination. Furthermore, the expertise of advisory committees on how to determine national priority groups for influenza vaccination was leveraged to determine the priority risk groups for COVID-19 vaccination.

### 3.4. National Technical Advisory Groups

The role of National Immunization Technical Advisory Groups (NITAG) as key experts responsible for providing independent, transparent, evidence-informed and timely recommendations on vaccines and immunizations to health authorities, was pivotal in guiding outbreak responses and aiding in COVID-19 pandemic recovery efforts. Most NITAGs in the Region served as the main advisory body guiding the immunization program on the introduction and use of COVID-19 vaccines. This responsibility included the definition of groups at higher risk for severe illness (or complications) who should be prioritized for vaccination [29]. Operational recommendations for reaching these groups also benefitted from the existing national guidance for influenza vaccination [14,16].

### 3.5. Experience in Vaccination Campaigns

Health staff already familiar with mass vaccination campaigns for both young children and older adults accelerated the deployment of COVID-19 vaccines. Their experience administering vaccines to multiple age groups facilitated the rapid scale-up of COVID-19 vaccination campaigns on a massive scale and under considerable political and societal pressure. Countries with a seasonal influenza vaccination program reported fewer problems with cold chain capacity (7%) compared to countries with no vaccination program (25%) [14]. Also, experience with influenza vaccination campaigns allowed COVID-19 vaccination events to be tailored to the needs of priority groups (e.g., convenient timing and location of the vaccination sites; transportation; accommodations while waiting). Public awareness and education campaigns on influenza vaccination—targeting both the general population and high-risk priority groups—served as a model for COVID-19 vaccination campaigns, helping to increase acceptance, uptake and coverage [14,16].

### 3.6. Vaccination Information Systems

The vaccination monitoring and recording systems developed for influenza—especially for the pediatric population—were adapted to track the distribution and administration of COVID-19 vaccines. In fact, some countries without pre-existing systems took advantage of the COVID-19 vaccine introduction to develop and implement nominal electronic immunization registries, use SMS reminders, conduct real-time data collection and reporting, and generate digital vaccination certificates; many of which are now being expanded to include other vaccines in their national immunization program (e.g., influenza, RSV). The introduction of influenza vaccines against influenza pandemic A(H1N1)pdm09 in 2009 and against COVID-19 highlighted the importance of monitoring and reporting adverse events following immunization, especially for new vaccines. PAHO has revised and updated the regional manual for the surveillance of Events Supposedly Attributable to Vaccination or Immunization (ESAVI) and established a regional strategy for conducting permanent and functioning ESAVI surveillance for all vaccines [30].

## 4. Influenza Vaccination as Part of the Life-Course Approach

The life-course approach to health provides a framework to understand the health and well-being of individuals and populations as the sum of capacities that are built, sustained, and recovered across both life stages and generations. Today, the life-course approach to immunization states that people should receive all recommended doses of vaccines throughout their lives to obtain the maximum benefits of protection against vaccine-preventable diseases at different ages, across generations, and in their communities [31,32]. Integrating vaccination programs for respiratory viruses into routine health services, such as adult scheduled immunization visits or antenatal care visits, can contribute to the lasting sustainability of vaccination programs at large. Below, we report examples of PAHO strategies and operations that contributed to strengthening national political and programmatic frameworks around life-course immunization using the influenza vaccination platform as a starting point.

### 4.1. Policy Recommendations from the WHO and PAHO Technical Advisory Groups

Since 2004, PAHO’s Technical Advisory Group (TAG) and WHO’s Strategic Advisory Group of Experts (SAGE) on immunization have built upon each other’s recommendations. In the context of COVID-19, the latest recommendations promote prioritization of high-risk groups (i.e., older adults, pregnant persons, persons with comorbidities, immunocompromised persons, health workers) for vaccination against respiratory infections [33,34,35,36]. Since 2021, both groups have recommended that the influenza vaccination platform be used to promote the rapid deployment and uptake of COVID-19 vaccinations, thus reducing the rates of severe disease, hospitalization, and death among priority populations. Additionally, coadministration of both influenza and COVID-19 vaccines during the same vaccination encounter, as recommended by WHO SAGE in October 2021 [37], could facilitate the implementation of integrated vaccine programmes, improve vaccination uptake, and decrease the overall burden on health services [38].

In November 2023, the TAG issued a statement where it strongly supported the SAGE recommendation that countries should integrate COVID-19 vaccination operations into their national immunization programs to maintain focus on achieving high vaccination coverage against the SARS-CoV-2 virus among priority groups and promote vaccination among all age groups [39]. During this same meeting, the TAG considered whether to recommend RSV vaccination among pregnant persons at 32–36 weeks of gestation to prevent disease in infants while minimizing the risk of preterm birth. The TAG did not issue comments on the coadministration of this vaccine with other antigens during pregnancy, but the US Centers for Disease Control and Prevention (CDC) states that the “RSVpreF vaccine can be administered to pregnant persons with other recommended vaccines, such as tetanus, diphtheria, and pertussis (Tdap), influenza, and COVID-19 vaccines” [40].

### 4.2. Addition of RSV to the SARInet Network

SARI*net* plus has long integrated RSV into national sentinel surveillance strategies, proving essential for providing comprehensive monitoring of virologic and epidemiological data to increase situational awareness and guide preventive measures [2,5]. The network improves coordination among national respiratory surveillance systems in both outpatient and hospital settings and incorporates RSV diagnostic testing at National Influenza Centers (NICs) and National Reference Laboratories (NRLs) contributing to the WHO Global Influenza Surveillance and Response System (GISRS).

SARI*net* plus supports public health decisions aimed at reducing RSV-related morbidity and mortality by analyzing trends, seasonal variations, and regional differences overall and in relation to the disease severity (i.e., RSV-related hospitalizations), focusing on high-risk groups like infants and older adults. Analyses dating back to 2010 have revealed significant regional variations in RSV seasonality, underscoring the need for tailored interventions [41]. Regional and climatic factors are known to influence RSV activity, typically starting in the south and moving northward with differing onset and peak times [42,43]. Understanding these patterns is crucial for optimizing interventions and mitigating the substantial burden of RSV across all age groups.

### 4.3. Latin American Center for Perinatology, Women and Reproductive Health (CLAP/WR) Network

Since 1983, CLAP/WR has promoted a standardized clinical record for maternal and perinatal care, known as the Perinatal Clinical Record, which includes information about vaccines administered during pregnancy [44]. This record is supported by the Perinatal Information System (SIP), available free of charge to all countries in the Americas. Nineteen health facilities in nine countries that are members of the “CLAP Network for Maternal Health Surveillance” regularly contribute their data to the Perinatal Clinical Record. The network maintains high-quality data on coverage with vaccines recommended for routine use during pregnancy, including COVID-19 vaccines. This platform has enabled the network to follow trends in vaccination coverage among pregnant persons, identify factors associated with the vaccine uptake and to monitor ESAVIs associated with various COVID-19 vaccination platforms used in these countries. More recently, the network has begun monitoring ESAVIs associated with RSV vaccination of pregnant persons in Argentina and Uruguay.

### 4.4. Behavioral and Social Drivers Studies

Behavioral and Social Drivers (BeSD) of vaccination are defined as beliefs and experiences specific to vaccination that are potentially modifiable to increase vaccine uptake [45]. The WHO developed the BeSD tool to help countries understand the reasons for low vaccine uptake, track trends over time, and reduce coverage inequities by gathering and using data to systematically design, implement, and evaluate tailored interventions [46]. In 2021 and 2022, PAHO used this methodology in countries of the Caribbean and Latin Americas to assess attitudes towards COVID-19 vaccines among health workers. By the end of 2024, PAHO will have implemented the Spanish version of the tool to collect perceptions around vaccination against influenza, COVID-19 and RSV among pregnant persons aged 18 years or older in Argentina. BeSD results can be used to inform the design and implementation of interventions that increase acceptance and confidence in vaccines in the persons at the highest risk of infection and severe disease.

Also, in 2024, PAHO published a performance monitoring tool to support countries in conducting a self-assessment of their national immunization program [47]. Results from these exercises will provide additional information about member states’ efforts and challenges in providing vaccination services to high-risk priority groups.

## 5. Discussion

The strong seasonal influenza vaccination program in the Americas, built over decades, prevents hundreds of thousands of illnesses, hospitalizations and influenza-associated deaths every year [48]. Following the 2009 A(H1N1) influenza and 2020–2023 COVID-19 pandemics, countries of the Americas have continued their efforts to sustain seasonal influenza vaccine uptake among priority groups at higher risk for severe illness (or complications), especially in the adult population and among pregnant persons. Countries have also continued strengthening their influenza surveillance systems, immunization platforms, and information systems—efforts which indirectly bolster preparedness for future pandemics [6,9,49]. Furthermore, influenza vaccination efforts have greatly contributed to the successful expansion of life-course immunization beyond childhood vaccines. It has facilitated the introduction of COVID-19 vaccines in the Americas and progressively reduced mortality due to vaccine-preventable diseases. Below, we outline six key recommendations that could support other WHO Regions in strengthening their seasonal influenza vaccination strategies.
Promote high-level advocacy and collaboration between Ministries of Health, PAHO Immunization Program and Revolving Fund for Access to Vaccines, and their partners to promote and consistently finance seasonal influenza vaccination for the general population and among high-risk priority groups.Strengthen technical cooperation between these stakeholders to develop updated guidance and tools and to strengthen national capacities for timely vaccination in priority groups. These materials supported member states in their decision-making to strengthen national policies and programs for influenza vaccination and improved the reach and quality of their seasonal programs [50].Expand the seasonal influenza vaccination program to persons aged 6 months or older, with a focus on young children, older adults, pregnant persons, people with comorbidities and health workers. Such an approach of vaccinating multiple age groups concomitantly allowed countries to implement and promote the concept of immunization across the life course well before it was included as a strategic priority of the Immunization Agenda 2030. As more vaccines come through the pipeline to address more diseases and conditions, these lessons learned about how to facilitate vaccine acceptance and delivery across the life course will become more and more critical.Document lessons learned from the introduction and roll-out of the COVID-19 vaccines during the pandemic. Employing existing structures of the seasonal influenza vaccination programs—such as regulatory pathways, vaccination strategies, campaign planning and implementation, cold chain capacities, information systems, public awareness and trained health workers—was essential for a rapid introduction and roll-out of COVID-19 vaccines. Having a strong seasonal influenza vaccination program contributes to better preparedness and would support emergency response operations during the next pandemic.Improve the completeness, timeliness and accuracy of the data flow after vaccines are in use in populations—from needs assessments to data collection, analysis, interpretation and interoperability. This step can strengthen the completeness, timeliness and reliability of the information on influenza vaccination programs and therefore promote the use of evidence to inform policy decisions.Intensify maternal vaccination operations, which can benefit from the existing capacities of the influenza vaccination program to reach this population. For pregnant persons specifically, the option of receiving these additional vaccines in conjunction with the annual dose against influenza or RSV can facilitate acceptance and uptake. The integration of vaccination operations with other essential care and antenatal services is likely to further boost coverage rates.

As part of its ongoing efforts to support immunization operations and reduce the burden of vaccine-preventable diseases, the Americas will continue to improve national immunization programs against respiratory pathogens and contribute to the lasting sustainability of vaccination programs at large.

## 6. Conclusions

Influenza vaccination protects people at risk from developing severe disease. It reduces the burden of disease, including hospitalization and deaths. It also protects health workers by reducing transmission and helps maintain the health system during influenza epidemics and pandemics. Seasonal influenza vaccination provides a sustainable platform for life course immunization against influenza and other vaccine-preventable diseases and contributes to a foundation for pandemic preparedness and response.

## Figures and Tables

**Figure 1 vaccines-12-01415-f001:**
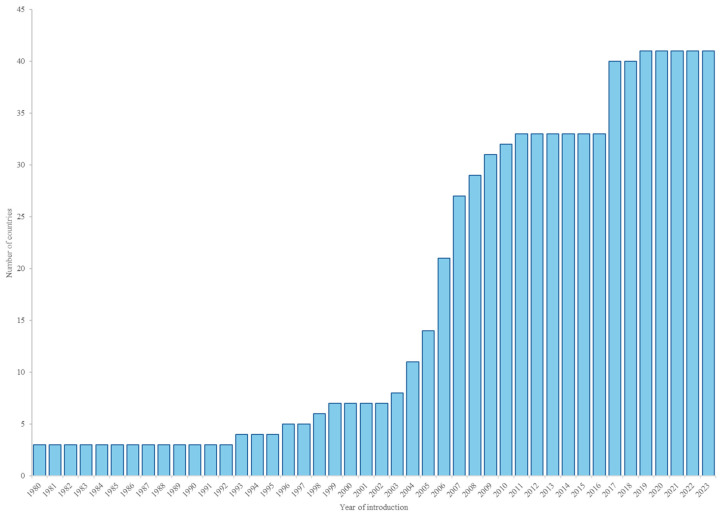
Cumulative number of countries with seasonal influenza vaccination by year, Region of the Americas, 1980–2023. Source: eJRF.

**Table 1 vaccines-12-01415-t001:** Seasonal influenza vaccination in the Americas, year of introduction, vaccine formulation and priority groups by country, 2023 (*n* = 44).

Country/Territory ^†^	Seasonal Influenza Vaccination * (*n* = 44)	Year of Introduction	Vaccine Formulation ^&^	Priority Groups ±				
				HCW °	Older Adults	Pregnant Persons	Persons with Chronic Diseases	Children
North America								
Canada	•	1980	NH	•	•	•	•	•
Mexico	•	2004	NH	•	•	•	•	•
United States of America	•	1980	NH	•	•	•	•	•
Caribbean								
Anguilla	•	2017	NH	•	•			
Antigua and Barbuda	•	2007	NH	•	•	•	•	•
Aruba	•	2017	NH	•	•	•	•	•
Bahamas	•	2005	NH	•	•	•	•	•
Barbados	•	2006	NH	•				
Belize	•	2008	NH	•	•	•		•
Bermuda	•	2017	NH	•	•	•	•	•
British Virgin Islands	•	2017	NH	•	•	•	•	
Cayman Islands	•	2017	NH	•	•	•	•	•
Cuba	•	1998	SH	•	•	•	•	•
Curaçao	•	2006	NH		•			
Dominica	•	2011	NH	•	•	•	•	•
Dominican Republic	•	2006	NH	•	•	•	•	•
Grenada	•	2006	NH	•	•	•	•	•
Guyana								
Haiti								
Jamaica	•	2006	NH	•	•	•	•	•
Montserrat	•	2017	NH	•	•			
Saint Kitts and Nevis	•	2019	NH	•	•	•	•	
Saint Lucia	•	2006	NH	•	•	•	•	
Saint Vincent and the Grenadines								
Sint Maarten	•	2009	NH	•	•	•	•	
Suriname	•	2009	NH	•	•		•	
Trinidad and Tobago	•	2007	NH	•	•	•	•	•
Turks and Caicos Islands	•	2017	NH	•	•	•	•	•
Central America								
Costa Rica	•	2004	SH	•	•	•	•	•
El Salvador	•	2004	SH	•	•	•	•	•
Guatemala	•	2007	NH	•	•	•	•	•
Honduras	•	2003	SH	•	•	•	•	•
Nicaragua	•	2007	SH	•	•	•	•	
Panama	•	2005	SH	•	•	•	•	•
Andean								
Bolivia	•	2010	SH	•	•	•	•	•
Colombia	•	2007	SH	•	•	•	•	•
Ecuador	•	2006	NH	•	•	•	•	•
Peru	•	2008	SH	•	•	•	•	•
Venezuela	•	2007	NH	•	•		•	
Southern Cone								
Argentina	•	1993	SH	•	•	•	•	•
Brazil	•	1999	SH	•	•	•	•	•
Chile	•	1980	SH	•	•	•	•	•
Paraguay	•	2005	SH	•	•	•	•	•
Uruguay	•	1996	SH	•	•	•	•	•
Number of countries—n (%)	41 (93%)	41 (93%)	41 (93%)	40 (98%)	40 (98%)	35 (85%)	36 (88%)	30 (73%)

Each grey cell represents the exclusion of the population group from the list of priority persons in each country to whom influenza vaccination should be offered. † Countries and territories reporting to PAHO/WHO. Additional seven countries/territories in the Americas Region do not report to PAHO/WHO: Bonaire, French Guiana, Guadeloupe, Martinique, Puerto Rico, Saba, and St. Eustatius. * In at least one priority group, either private or public sector. ^&^ Vaccine formulation NH: Northern Hemisphere; SH: Southern Hemisphere. ± Target groups and eligible age groups for vaccination (e.g., older adults, people with comorbidities and children) vary across countries. Percentage calculated based on 41 countries with influenza vaccination. ° HCW = health workers.

**Table 2 vaccines-12-01415-t002:** Regional median influenza vaccination coverage reported by priority groups in the Americas, 2023.

Priority Groups *	Countries/Territories Targeting The Priority Groups	Countries/Territories Reporting to PAHO eJRF °	Median Vaccination Coverage	Interquartile Range (IQR)
Health workers	40	18	61%	34–99%
Older adults	40	17	56%	15–74%
Pregnant persons	35	19	60%	40–81%
Persons with chronic diseases	36	11	85%	43–100%
Children aged ≥6–12 months †	30	19	47%	26–70%
Children aged 12–23 months †	30	15	72%	41–91%

Source: PAHO/UNICEF eJRF 2023. † Vaccinated with two doses or a single dose if previously vaccinated with two doses. * Different target populations and age groups are considered by the countries. ° eJRF = WHO/UNICEF electronic Joint Reporting Form.

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
