# Peer review of "Seasonal Influenza Vaccination Programs in the Americas: A Platform for Sustainable Life-Course Immunization and Its Role for Pandemic Preparedness and Response"

_vaccines, 2024, doi:10.3390/vaccines12121415_

Round 1
Reviewer 1 Report
Comments and Suggestions for Authors
The study is engaging and holds significant interest; however, it lacks key components expected in a research article. So it should be rejected Since it is submitted as a research article, it should include distinct sections for Methodology, Results, and Discussion. Currently, the manuscript transitions from the Introduction and other sections with various subtitles directly to the Conclusion, bypassing these critical sections. Additionally, the references are not formatted according to the journal's guidelines. Adhering to the journal's requirements is essential for maintaining academic rigor and ensuring proper presentation of the study. Furthermore, the references were not formatted according to the MDPI journal guidelines. It is essential to revise the references to align with the required MDPI format.
Author Response
The study is engaging and holds significant interest; however, it lacks key components expected in a research article. So it should be rejected Since it is submitted as a research article, it should include distinct sections for Methodology, Results, and Discussion. Currently, the manuscript transitions from the Introduction and other sections with various subtitles directly to the Conclusion, bypassing these critical sections. Additionally, the references are not formatted according to the journal's guidelines. Adhering to the journal's requirements is essential for maintaining academic rigor and ensuring proper presentation of the study. Furthermore, the references were not formatted according to the MDPI journal guidelines. It is essential to revise the references to align with the required MDPI format.
Answer: Thank you for this positive feedback and for flagging the type and format of this manuscript. Indeed, this manuscript is not a research article, but rather a review of the influenza vaccination in the Americas. The manuscript will be re-submitted as a Review. In addition, we have updated and formatted the references according to the MDPI journal guidelines.
Reviewer 2 Report
Comments and Suggestions for Authors
-
Dear Editor,
The article titled “Progress of the seasonal influenza vaccination programs in the Americas and its role during the introduction of the COVID-19 vaccines” provides the reader with an overview of seasonal influenza vaccination in the Americas over the past 50 years, highlighting both the achievements and challenges during this period. The manuscript, submitted as a review, is well-structured and addresses various aspects of the processes and policies related to vaccination campaigns during this time. It also discusses the positive impact of countries with well-established influenza vaccination surveillance systems on their response to the COVID-19 pandemic.
Below are minor revisions that may help improve the organization and clarity of the manuscript.
- P2L71 – Please, specify what platforms?
- P2L75 – Was the United States the first country to apply the seasonal influenza vaccine in the Americas?
- P2L84 – Shouldn't the authors specify those countries with well-established seasonal influenza vaccination programs and better preparedness during the COVID-19 pandemic?
- P3L102 – Between 1945 and 1980, was the vaccine not introduced in any other country in the Americas? The sentence states that seasonal influenza vaccination was introduced in the U.S. in 1945, but Table 1 indicates it was in 1980.
- Tab 1 – For "Older Adults," the authors should specify the minimum age considered (e.g., > 65 years).
- P5L124 – It is important that the authors define for the reader what the Northern and Southern Hemisphere vaccine formulations are and explain the differences between them.
- P5L130 – What is the trivalent vaccine? What is it composed of?
- P6L173 – The authors should specify the meaning of eJRF.
- Tab 2 – Table 2 should be accompanied by a figure showing a map of the Americas with influenza vaccination coverage stratified by countries over the past five years, including the pandemic period (2020–2022).
- P7L212 – Vaccine effectiveness: How is it evaluated? It would be worthwhile for the authors to briefly explain how influenza vaccine effectiveness is estimated by national immunization programs without using controlled epidemiological studies (clinical trials or case-control studies).
Author Response
The article titled “Progress of the seasonal influenza vaccination programs in the Americas and its role during the introduction of the COVID-19 vaccines” provides the reader with an overview of seasonal influenza vaccination in the Americas over the past 50 years, highlighting both the achievements and challenges during this period. The manuscript, submitted as a review, is well-structured and addresses various aspects of the processes and policies related to vaccination campaigns during this time. It also discusses the positive impact of countries with well-established influenza vaccination surveillance systems on their response to the COVID-19 pandemic.
Thank you very much for the revision and for the comments provided. Please see below the answers to your suggestions.
Below are minor revisions that may help improve the organization and clarity of the manuscript.
P2L71 – Please, specify what platforms? – Clarified in the text
P2L75 – Was the United States the first country to apply the seasonal influenza vaccine in the Americas? – Yes, US was the first country in the region to introduce the vaccine. Indicated in line 101.
P2L84 – Shouldn't the authors specify those countries with well-established seasonal influenza vaccination programs and better preparedness during the COVID-19 pandemic? - We preferred not mentioning countries.
P3L102 – Between 1945 and 1980, was the vaccine not introduced in any other country in the Americas? The sentence states that seasonal influenza vaccination was introduced in the U.S. in 1945, but Table 1 indicates it was in 1980. – The table starts from 1980 because we have data available from this year. We know that US introduced the vaccine in 1945, but not information from other countries until eJRF data available.
Tab 1 – For "Older Adults," the authors should specify the minimum age considered (e.g., > 65 years). – Indicated and specified in the manuscript.
P5L124 – It is important that the authors define for the reader what the Northern and Southern Hemisphere vaccine formulations are and explain the differences between them. – Clarified in the text – line 123.
P5L130 – What is the trivalent vaccine? What is it composed of? – included in the text.
P6L173 – The authors should specify the meaning of eJRF. – indicated in line 105.
Tab 2 – Table 2 should be accompanied by a figure showing a map of the Americas with influenza vaccination coverage stratified by countries over the past five years, including the pandemic period (2020–2022). – Vaccination coverage data is presented annually at the PAHO Immunization Brochure. We have included a reference to this report.
P7L212 – Vaccine effectiveness: How is it evaluated? It would be worthwhile for the authors to briefly explain how influenza vaccine effectiveness is estimated by national immunization programs without using controlled epidemiological studies (clinical trials or case-control studies). – Included brief explanation in the text.
Reviewer 3 Report
Comments and Suggestions for Authors
General:
Define WHO and PAHO across the manuscript
Stick and follow the journal reference numbering as per the journal style across the manuscript
Could you also provide on the role of: Strong research implemented and bodies responsible as well as other stakeholder and networking?
Abstract
Lines 44-47. This is the review objectives and I recommend to be upfront in the abstract
Introduction
Lines 54-55. Provide the reference
Lines 55-58. Shift the review aims to the end of the introduction that will give a good flow of the justification of the review. I believe in line 88, you have already provided the review objectives therefore you don’t have to repeat them in lines 55-58.
Comment: it will be interesting to provide some global context as well.
Influenza vaccine formulations
Lines 129-131. Provide the reference
Quality of influenza vaccination monitoring data
Lines 151-153 provide the reference
Author Response
General:
- Define WHO and PAHO across the manuscript – Revised, completed and manuscript updated.
- Stick and follow the journal reference numbering as per the journal style across the manuscript – References have been updated according to MDPI journal guidelines.
- Could you also provide on the role of: Strong research implemented and bodies responsible as well as other stakeholder and networking? – Information included in the text.
Abstract:
- Lines 44-47. This is the review’s objectives, and I recommend to be upfront in the abstract – The objective of this Review has been moved up in the abstract.
Introduction:
- Lines 54-55. Provide the reference – Reference included.
- Lines 55-58. Shift the review aims to the end of the introduction that will give a good flow of the justification of the review. I believe in line 88, you have already provided the review objectives therefore you don’t have to repeat them in lines 55-58. – Revised and manuscript updated.
Comment: it will be interesting to provide some global context as well – Global context included in the introduction.
Influenza vaccine formulations
- Lines 129-131. Provide the reference – Reference included.
Quality of influenza vaccination monitoring data
- Lines 151-153 provide the reference – Reference included.
Thank you very much for your valuable comments. We have updated the manuscript and abstract following your comments and suggestions. We included additional references as indicated.
Reviewer 4 Report
Comments and Suggestions for Authors
This is an important piece of work that can, as authors described, serve as a blueprint for the introduction of new vaccines into wider practice. Here are some of my comments and suggestions to consider for improving the paper.
Title “Progress of the seasonal influenza vaccination programs in the Americas and its role during the introduction of the COVID-19 vaccines” seems a bit inappropriate since this is not timeline of the evolution of programs (beside the years of introduction into practice) but rather an overview of situation, please consider modifying it to better reflect the aims, results and conclusions of this paper. Also, it should include a statement that it is a “review” and not original article.
Sentence in line 88 needs to add “current” seasonal influenza vaccination program of the Americas.
References should be properly formatted in the text (not using roman numbers)
Table 1: Older adults - is it uniformly presented across the countries (above 65yrs old?) - needs to be specified
Persons with chronic diseases - is it uniformly presented across the countries which chronic diseases - needs to be specified
Children - is it uniformly presented across the countries? 6months to 18 yrs old or else? should be specified
sum for the second column (Seasonal influenza vaccination) is listed as 44 (100%) but 3 countries don’t have a program, so it should be also 41 (93%) as the next one. SIdenote suggestion, this column can also be removed since in the next one is clear if no year is present that there is no Seasonal influenza vaccination.
It remains unclear (needs to be specified) that Priority groups are uptodate (in current programe) and not the same since the introduction, since I assume these programs evolved over time (including more and more groups)?
This table is the key to this work, perhaps consider grouping countries in some other way (beside just alphabetically?), since countries across the Americas are much diverse, in many ways and especially economically. Since this fact very much influence the availability of healthcare service as well as disease outcome perhaps try grouping country in subsection based on GDP or somehow to better emphasizes these differences. Many times this monetary issue shapes the way priority groups are defined (and not just health risk)
Fig1 needs to be explained in the text. It seems it is a cumulative number for each year counting for all countries? this needs to be specified. Also, I would say perhaps this information is a bit redundant since it is also presented in the Tavle1?
I find Fig2 to be not much useful, so perhaps consider presenting this data in some other form. I mean, it is expected that countries above the Equator use North hemisphere and below use Southern formulation, only those few countries closest to Equator are interesting since can go both ways - how they decide? Why is this? how to decide which programme to use? considering the recommendations from the WHO based on the influenza types circulating or something else?
Wouldn’t it be more useful to have this data in Table 2 “vaccination coverage reported by priority group in the Americas, 2023”, presented per country/territory? If it is too much data consider placing it in the Supplement? I mean, main point here that if we analyse quality programs that are different across different countries we would need to see the effects of that programs, i.e., the reported coverage by country.
Statements made in the line 270 stating that those countries that had influenza pogrames responded better to COVID-19 , i.e., “Countries with robust influenza vaccination programs were able to define these groups more quickly and precisely based on their prior experience [ref 33]” need to have it documented, how was this evaluated? do you have this data as well? I mean, we would need to have objective evaluation of this by presenting the data for this statement, since this is one of the aims of this review paper (i.e., “as well as examine how the program contributed to a timely roll-out of 90 COVID-19 vaccines during the recent pandemic”). On the other hand, if this is a conclusion from some other study, this needs to be clearly stated and not presented as a result of this work [not just by reference, since we don’t know if they concluded it or you refer to their data]
Finally, statements like this “countries with existing seasonal influenza vaccination programs were better prepared and rolled out the delivery of COVID-19 vaccines more quickly and effectively compared to other countries where the influenza vaccination platform was weak or non-existent.” need objective evaluation and be support by the data (I’m not saying that are not, just that we need to have it presented here across the paper).
lines 275-284 need references
Conclusions presented in lines 401-437 are useful but are not a result of the data presented in this paper. I would rather see this as recommendations than conclusion section. Conclusions on the other hand should be a brief summary of the presented results in the paper.
Author Response
This is an important piece of work that can, as authors described, serve as a blueprint for the introduction of new vaccines into wider practice. Here are some of my comments and suggestions to consider for improving the paper.
Thank you for your review and for your positive feedback.
- Title “Progress of the seasonal influenza vaccination programs in the Americas and its role during the introduction of the COVID-19 vaccines” seems a bit inappropriate since this is not timeline of the evolution of programs (beside the years of introduction into practice) but rather an overview of situation, please consider modifying it to better reflect the aims, results and conclusions of this paper. Also, it should include a statement that it is a “review” and not original article. - We have updated the title of the manuscript to better reflect the content of the information presented. As pointed out by the Reviewer #1, this is not a research article, but rather a review. We will re-submit the manuscript as a Review.
- Sentence in line 88 needs to add “current” seasonal influenza vaccination program of the Americas - Corrected
- References should be properly formatted in the text (not using roman numbers) – References updated and formatted according to MDPI guidelines.
- Table 1: Older adults - is it uniformly presented across the countries (above 65yrs old?) - needs to be specified
- Persons with chronic diseases - is it uniformly presented across the countries which chronic diseases - needs to be specified
- Children - is it uniformly presented across the countries? 6months to 18 yrs old or else? should be specified. Target groups and eligible age groups for vaccination (e.g. older adults, people with comorbidities and children) vary across countries. Also, the priority groups have been expanded over time since the introduction of the vaccine. The data presented in table 1 refers to the current (2023) vaccination target groups. We have highlighted this in the manuscript.
- sum for the second column (Seasonal influenza vaccination) is listed as 44 (100%) but 3 countries don’t have a program, so it should be also 41 (93%) as the next one. SIdenote suggestion, this column can also be removed since in the next one is clear if no year is present that there is no Seasonal influenza vaccination. – Table updated with the suggestion. Included footnote to clarify the totals.
- It remains unclear (needs to be specified) that Priority groups are up-to-date (in current programe) and not the same since the introduction, since I assume these programs evolved over time (including more and more groups)? – This has been clarified and specified in the section “Influenza vaccine introduction” of the manuscript.
- This table is the key to this work, perhaps consider grouping countries in some other way (beside just alphabetically?), since countries across the Americas are much diverse, in many ways and especially economically. Since this fact very much influence the availability of healthcare service as well as disease outcome perhaps try grouping country in subsection based on GDP or somehow to better emphasizes these differences. Many times this monetary issue shapes the way priority groups are defined (and not just health risk). – We have grouped countries in table 1 geographically by sub-region (North America, Central America, Caribbean, Andean region and Southern Cone).
- Fig1 needs to be explained in the text. It seems it is a cumulative number for each year counting for all countries? this needs to be specified. Also, I would say perhaps this information is a bit redundant since it is also presented in the Tavle1? – This has been clarified in the text and in the figure 1.
- I find Fig2 to be not much useful, so perhaps consider presenting this data in some other form. I mean, it is expected that countries above the Equator use North hemisphere and below use Southern formulation, only those few countries closest to Equator are interesting since can go both ways - how they decide? Why is this? how to decide which programme to use? considering the recommendations from the WHO based on the influenza types circulating or something else? – We have included this information in table 1 with a new column “Vaccine formulation” and removed figure 2.
- Wouldn’t it be more useful to have this data in Table 2 “vaccination coverage reported by priority group in the Americas, 2023”, presented per country/territory? If it is too much data consider placing it in the Supplement? I mean, main point here that if we analyse quality programs that are different across different countries we would need to see the effects of that programs, i.e., the reported coverage by country. We did not present in this manuscript the coverage by country and by priority group because this data is already published in the PAHO Immunization Brochure. The reference to the PAHO Immunization Brochure is included in the manuscript in section “Seasonal influenza vaccination coverage by priority group”.
- Statements made in the line 270 stating that those countries that had influenza pogrames responded better to COVID-19 , i.e., “Countries with robust influenza vaccination programs were able to define these groups more quickly and precisely based on their prior experience [ref 33]” need to have it documented, how was this evaluated? do you have this data as well? I mean, we would need to have objective evaluation of this by presenting the data for this statement, since this is one of the aims of this review paper (i.e., “as well as examine how the program contributed to a timely roll-out of 90 COVID-19 vaccines during the recent pandemic”). On the other hand, if this is a conclusion from some other study, this needs to be clearly stated and not presented as a result of this work [not just by reference, since we don’t know if they concluded it or you refer to their data]. Finally, statements like this “countries with existing seasonal influenza vaccination programs were better prepared and rolled out the delivery of COVID-19 vaccines more quickly and effectively compared to other countries where the influenza vaccination platform was weak or non-existent.” need objective evaluation and be support by the data (I’m not saying that are not, just that we need to have it presented here across the paper). – We have included data in the text and references that supports this statement.
- lines 275-284 need references – References included in the text.
- Conclusions presented in lines 401-437 are useful but are not a result of the data presented in this paper. I would rather see this as recommendations than conclusion section. Conclusions on the other hand should be a brief summary of the presented results in the paper. - The conclusion section has been updated as a discussion and recommendation points.
Round 2
Reviewer 1 Report
Comments and Suggestions for Authors
Good
Reviewer 3 Report
Comments and Suggestions for Authors
NIL
Reviewer 4 Report
Comments and Suggestions for Authors
Thank you for making modifications to the manuscript and for providing responses to my concerns.